# Comparison of Capillary Flow Porometry (CFP) and Liquid Extrusion Porometry (LEP) Techniques for the Characterization of Porous and Face Mask Membranes

**R. I. Peinador [1,\*], José I. Calvo [2,3] and Roger Ben Aim [1]**

[1] Institut de la Filtration et des Techniques Séparatives (IFTS), Rue Marcel Pagnol,
47510 Foulayronnes, France, roger.benaim@ifts-sls.com

[2] Departamento de Física Aplicada, ETSIIAA, University of Valladolid, 34071 Palencia, Spain

[3] Surfaces and Porous Materials (SMAP), Associated Research Unit to CSIC, UVainnova bldg,
Pº de Belén 11 and Institute of Sustainable Processes (ISP), Dr. Mergelina s/n, University of Valladolid,
47071 Valladolid, Spain, joseignacio.calvo@uva.es

\* Correspondence: rene.peinador@ifts-sls.com

**Featured Application: Membrane characterization**

**Abstract:** This work aims to study the characterization of several membrane filters by using capillary flow porometry (CFP) and liquid extrusion porometry (LEP) to obtain their pore size distributions (PSD) and mean pore diameters ($d_{avg}$). Three polymeric membranes of different materials namely, polyethylene (PET), cellulose nitrate (CN), and FM (face mask), and one inorganic (namely, alumina $Al_2O_3$) from ultrafiltration (UF)/microfiltration (MF) and particle separation were analyzed using a pressure constant fluid/liquid extrusion porometer, developed at institute de la filtration et techniques séparatives (IFTS). Several porosimetric fluids have been used to wet and penetrate into the porous/fiber structure. The results show the accuracy of the setup on characterizing membranes in the UF/MF range by CFP, with reasonable agreement with nominal data of the filters. Additionally, LEP extension of the equipment obtained good agreement with nominal data and the CFP results, while filters presenting a microstructure of highly interconnected pores (face mask) resulted in clear differences in terms of resulting PSD and average sizes when CFP and LEP results are compared.

**Keywords:** membrane characterization; pore-size distribution (PSD); capillary pressure; CFP; LEP

---

## 1. Introduction

Characterization of the pore-size distribution (PSD) of porous/fiber materials has been widely studied among the scientific community and membrane manufacturers [1,2]. Understanding the constitution, structure, and functional behavior of the membrane has improved in the last few years as the membrane properties and the membrane process has been scaled up, coupled with the economic large-scale manufacture of the membrane [3]. This knowledge needs to be acquired by using appropriate methods, because the studied membranes are put into a large number of different uses and application, even within a particular separation process, then a membrane will be characterized in terms of its pore size, molecular weight cut-off, (MWCO), porosity, thickness, symmetry, permeability, hydrophobicity and hydrophilicity, adsorption, crystallization, etc. In other words, we can see that a complete characterization should include both the functional aspects of the

problem and the structural ones, which are no less important. The characterization of pore size includes [4] the determination of (average) pore size diameter and the pore size distribution (PSD). The average pore size of the membrane filter can determine, in the first instance, the size of the molecules to be retained. However, more complete knowledge would be acquired if we supply not only the average or mean pore size, but also the complete PSD, for example, the amount or percentage for each pore size of those present in the membrane. Particle removal performance can be simply analyzed by using the PSD trends of a membrane [5]. There are two main classification groups for membrane pore size characterization methods, divided into direct and indirect methods, respectively. The direct methods include all the microscopic techniques (e.g., scanning electron microscopy (SEM), transmission electron microscopy (TEM), atomic force microscopy (AFM), scanning tunneling microscopy (STM), which are only able to directly measure the pores visible in the pictures) and spectroscopic techniques (e.g., electrical impedance spectroscopy, positron annihilation spectroscopy, raman spectroscopy, nuclear magnetic resonance, which are more useful to directly visualize the membrane material) [6–9]. On the other hand, indirect methods are based on some more or less complicated modeling to convert the direct data obtained for each filter into the resulting PSD. In this group, we can include techniques such as capillary flow porometry/liquid intrusion–extrusion techniques (CFP/LIEP), gas adsorption–desorption (GAD), evaporation techniques such as thermoporometry (THP), permoporometry (PMP), or the newest evapoporometry (EP) and solute rejection test. All these techniques can be used to determine the PSD of the porous membrane. The novelty of this study is in the comparison of the use of CFP and LEP indirect methods to characterize pore size distributions and mean pore diameters in a fully automated commercial setup available for performing both techniques. In this way, a single device is able to cover most of the filtration range from ultrafiltration (UF; 10 to 100 nm), microfiltration (MF; 0.1 to 10 µm) to particle filtration (PF; >10 µm) [10], reliably analyzing membranes with porous/fiber structure and made of quite different materials.

This work aimed to assess their respective benefits and drawbacks demonstrating the features, potential, and possible problems arising from the use of two quite simple, fast, and cheap techniques based on the use of the Young–Laplace equation for interpretation of the results.

## 2. Theory

### 2.1. Capillary Flow Porometry (CFP) Principles

The capillary flow porometry technique, also known as fluid–fluid displacement porometry (FDP), mainly comprises two similar techniques, gas liquid displacement porometry (GLDP), and liquid–liquid displacement porometry (LLDP) (Figure 1)

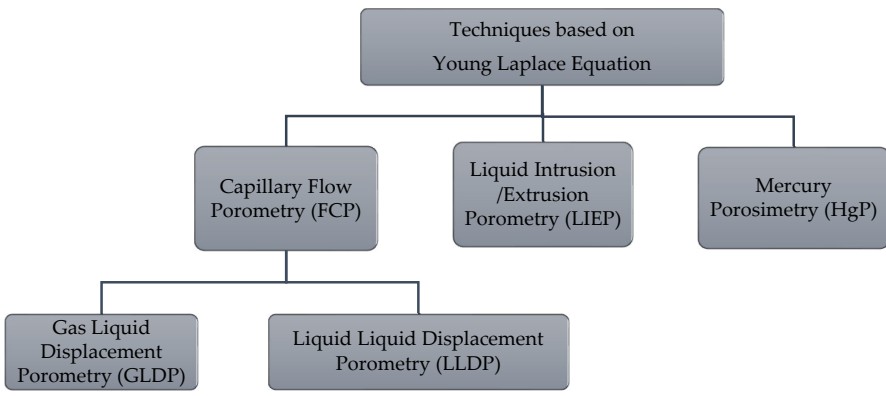

**Figure 1.** Young–Laplace-based techniques.

The CFP technique is based on the well-known Young–Laplace (Equation (1)) equation that governs the pressure difference at the interface between two immiscible fluids [11].

Gas liquid displacement porometry (GLDP) is a form a displacement technique in which both the differential gas pressure and the gas flow rates are simultaneously measured (first on a wet sample, followed by a similar measurement in a dry one). The technique has a typical lower size range of 50 nm for a maximum applied pressure around 14 bar, which restricts the application of the technique to the characterization of MF membranes or the upper limit of UF ones. To analyze membranes with pores smaller than 50 nm (as usually found in typical UF and all NF membranes), the air–liquid interface requires too much pressure, allowing for membrane structure damage. In this case, liquid–liquid displacement porosimetry (LLDP) circumvents this drawback by making use of a liquid–liquid interface [11] inside the pores. Then, the main difference between the two techniques is the use of a gas–liquid–solid versus a liquid–liquid–solid interface, the latter of which results in a substantial reduction of the surface tension of the corresponding interface and, accordingly, the necessity of using smaller pressures to analyze such small pores. As a result, the combination of GLDP and LLDP flow based techniques allows for the accurate characterization of membranes in across the entire range of MF and UF and, even, partially that of NF.

Both techniques (Figure 2) consist of a pressurized fluid (gas or liquid) that is introduced forcefully inside the pores of a pre-wetted (by means of a proper wetting liquid) filter medium. Operationally, the procedure consists of gradually increasing the applied pressure so that pores of decreasing sizes are opened in each step. When this fluid pressure overcomes the capillary intrusion pressure of the largest pore, the displacement fluid can penetrate into those pores and push out the wetting liquid. The pore size $d_p$ *(m)* of the pores open to flow at each applied pressure, $\Delta P$ *(Pa)*, is calculated by the Young–Laplace equation:

$$\Delta p = \frac{4\gamma cos\theta}{d_p} \tag{1}$$

where $\gamma$ *(n/m)* is the surface tension of the wetting liquid, and $\theta$ is the contact angle between the liquid and capillary wall. To obtain reliable results, the value of cos θ must be assumed to be equal to one (perfect wetting) for both techniques. The pressure at which the displacement fluid starts to flow through the pore is called a bubble point [12–14] (GLDP) or droplet point (LLDP). The pore size at the bubble/droplet point is termed as the maximum pore size of the sample filter. When the pressure is further increased, the displacement fluid flows through smaller pores.

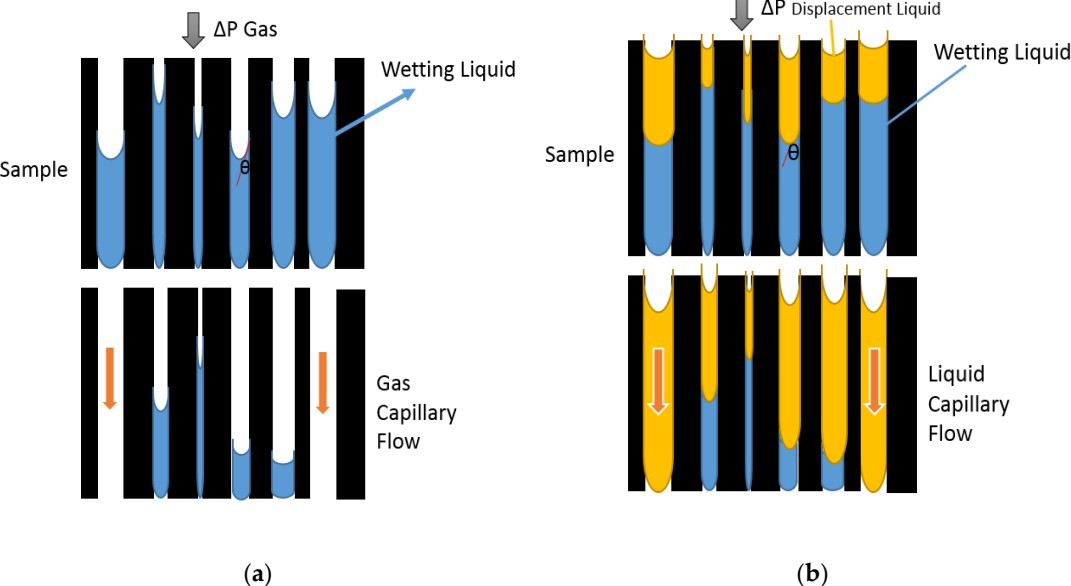

**(a)**          **(b)**

**Figure 2.** (**a**) Young–Laplace based techniques for gas liquid displacement; (**b**) liquid displacement.

## 2.2. Liquid Intrusion/Extrusion Porometry (LIEP) Principles

The intrusion/extrusion technique for pore size characterization on membrane media operates in similar way as the well-known mercury porometry (HgP) technique and, consequently, it is also among the characterization methods that are also based on the Young–Laplace equation [15–19]. The HgP technique relies on the fact that, in this case, it is mercury (i.e., a non-wetting liquid) that is forced to enter the pores measuring the intruded volume (more specifically, the differential increment of specific volume) versus applied pressure. These variables contain all the information needed to obtain the PSD of the sample. Normally, HgP runs include a complete intrusion–extrusion cycle. In the extrusion part of the cycle, the previously intruded mercury is ejected from the sample during the depressurization step.

Similarly, LEIP, which can also be considered as a variation of the GLDP technique, consists of wetting the membrane to be analyzed with an appropriate wetting fluid and then placing the membrane onto a capillary barrier membrane. Then, pressure of a highly hydrophilic liquid (intrusion liquid) is applied in such a way that both the pores of the sample and capillary membrane are steeply filled [20] (Figure 3). Both media are immersed into the wetting fluid.

To obtain a proper analysis, it must be assured that the capillary barrier membrane is such that its largest pore is smaller than the smallest pore of interest in the sample. Consequently, the gas pressure (i.e., non-wetting fluid) sufficient to extrude or drain the liquid from the pores of the sample is inadequate to empty the pores of the membrane acting as capillary barrier. Liquid extruded from the pores of the sample under gas pressure flows through the pores of the capillary barrier, while the pores of the membrane remain filled with the wetting liquid and prevent gas from passing through.

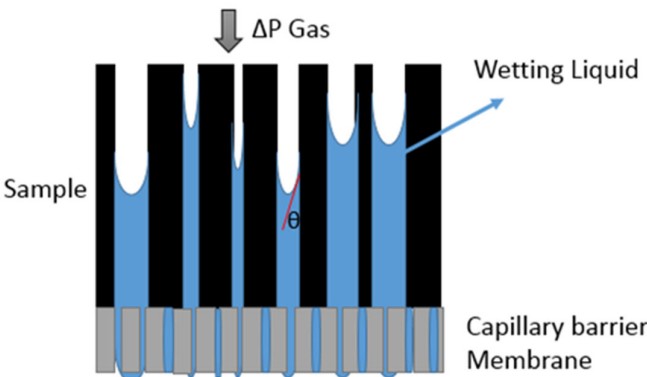

**Figure 3.** Principles of liquid extrusion porometry (LEP).

Liquid extrusion/intrusion porosimetry is a method by which the capillary pressure (i.e., difference pressure of the non-wetting and wetting liquid in the sample) is changed by directly controlling the gas pressure applied to the sample. The wetting liquid mass coming out from the sample is tracked by measuring the amount of liquid uptake on an analytical balance. As the system reaches equilibrium at each capillary pressure, the volume (mass) of liquid drained and the differential pressure are recorded prior to further pressure increases, [21–25]. After completion of an intrusion run, a capillary pressure curve can be generated. Once the maximum pressure has expelled the intrusion liquid from all the pores of the sample (i.e., there is no more contribution of the liquid drained), the capillary pressure is slowly relieved and the wetting liquid flows back into the sample. Using these draining and flowing back data, the capillary pressure curves (frequently showing some hysteresis) can be generated [20], and using the Young–Laplace equation, converted into pore size information. Quite often, only the extrusion run is used, and is not followed by a new intrusion experience, and then the method can be simplified as liquid extrusion porometry (LEP).

## 3. Materials and Methods

Four flat-sheet membranes supplied from recognized manufacturers were characterized and were selected so that the pore sizes spanned most of the UF–MF–Particle Filtration range, and also made of different materials, namely: alumina ($Al_2O_3$), polyethylene (PET), cellulose nitrate (CN), and a face mask (FM) filter made of layered cotton fibers integrated into surgical masks and respirators (Figure 4).

In the case of the PET and FM media, the membranes analyzed were still in a pre-commercialized stage, then under-going further research, so they were studied under a non-disclosure agreement that prevented the publishing of the company name (these are designated as Company A and Company B, respectively). The information available on the membranes and characterization method are given in Table 1.

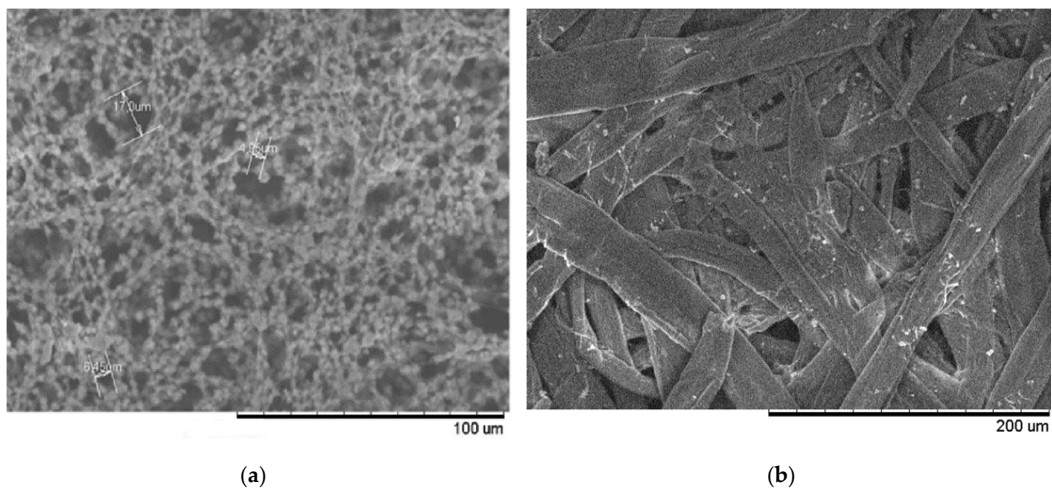

(**a**)　　　　　　　　　　　　　　　　　　　　(**b**)

**Figure 4.** SEM images of (**a**) NC 5μm porous structure and (**b**) FM fibers.

**Table 1.** Characteristics of the four membranes analyzed and porometric technique/s used.

| Material | Manufacturer | Pore Range | Nominal Pore Size | Type | Structure | Young-Laplace Technique |
|---|---|---|---|---|---|---|
| $Al_2O_3$ | Whatman | UF | 0.02 μm | Disk | Uniform capillary | LLDP |
| PET | Company A | UF | Not available | Flat Sheet | Porous | LLDP |
| CN | Sartorius | MF | 5 μm | Disk | Porous | GLDP/LEP |
| FM | Company B | Particle Filtration | Not available | Flat Sheet | Fiber | GLDP/LEP |

All membranes were obtained from each manufacturer in the form of flat disks with a diameter of 47 mm, except for the face mask ones that were available in the form of flat sheets. UF membranes were tested in a specific 47 mm diameter housing cell, while for the MF and particle filtration filters, a proper 25 mm diameter testing cell was used. All samples were dried out and immersed into the porosimetric wetting liquids for half an hour under vacuum pressure (200 mmHg) at room temperature to assure complete membrane soaking. Each membrane was tested using two different samples taken from the same batch. The results were averaged and the standard deviation of the results was calculated, and finally, the percentage of error was calculated as standard deviation/average value.

### 3.1. Capillary Flow Porometry (CFP) Method

The gas liquid and liquid displacement measurements were performed with IFTS fluid porometer (FFP) (model PRM-8710®, IFTS, France) consisted of an automated pressure constant device suitable for working in gas/liquid and liquid/liquid configurations. The device is designed for testing pore sizes down to 4 nm and up to 200 μm and uses relatively low pressures from 2 KPa (minimum) up to a maximum of 1 MPa for the characterization of porous or fiber membranes in the NF to the particle filtration range. The equipment allows for very stable pressure (accuracy ±0.1 mbar) to be implemented, which leads to very accurate measurement of the resulting fluxes by using an analytical balance (accuracy of ±1 mg) for LLDP characterization or by mass flowmeter (accuracy ±1 mL/min) for gas liquid. The included software is able to determine several important parameters related with pore size characterization including mean pore diameter, peak pore size, PSD, fluid permeability, and bubble/droplet point. It also can be adapted to analyze various membrane configurations or modules including hollow fiber, tubular, and flat sheet.

Capillary flow porometry (in both versions, gas liquid and liquid–liquid) was performed according to the capillary principle to obtain the PSD of the membranes. The filter media was previously wet in a liquid (wetting phase), then by increasing the pressure of the displacing fluid upstream to the membrane at a predetermined rate, the flow downstream of the membrane is observed, indicating the passage of displacement fluid (gas or liquid) through all the pores in the filter, starting at the maximum diameter filter (bubble point for gas liquid or "droplet point" for liquid–liquid) pores until flowing through the smallest ones. Both wetting and displacement fluids are immiscible between them.

For the case of UF membranes, LLDP experiments were performed according to the usual procedure,[19], and the wetting and displacement liquids used in LLDP analysis were selected according to the membrane hydrophobicity/hydrophilicity, good chemical compatibility with the material matrix active layer, and with the porometer internal parts to avoid corrosion or degradation. Wetting fluid should also exhibit physical properties such as low displacement viscosity and low vapor pressure. A very stable 1:1 (*v/v*) binary mixture composed of water/isobutanol ($\gamma$ =1.7 mN/m) was used. This mixture was prepared by pouring proper amounts of Milli-Q® grade water and isobutanol (methyl-2-propanol) into a separator funnel and shaking it vigorously during 10 min. After mixing, the two rapidly separating phases are then allowed to stand overnight and then two sharply and clearly separated phases are presented. The higher density phase and water-rich (aqueous) phase is first drained off and used as the displacement liquid, while the alcoholic phase (remaining in the upper part of the funnel due to their lower density) is the wetting one . Normally, the alcoholic phase due to better strong affinity with the mostly polymer part and inorganic media is chosen as the wetting liquid instead of the aqueous phase, which presents low viscosity, so ~0.001 Pa s was selected as the displacement liquid for the LLDP test.

The whole contribution from open pores to displacement liquid is calculated in terms of permeability, which is defined as the flux to pressure ratio and differential increase of permeability given by:

$$\Delta L_k = \frac{L_k - L_{k-1}}{L_{tot}} \tag{2}$$

where $L_k$ is the permeability of the *k-th step (k = 1, 2 i)* and $L_{tot}$ is the final permeability (asymptotic permeability) in the final *step* (i.e., *k = i)*. The value of the asymptotic permeability, $L_{tot}$, corresponds to the moment when the wetting liquid is drained away from all the membrane pores. The plot of contributions to total permeability for each pore size opened (then for each pressure increase step) can be understood as a pore size distribution in terms of flow (or permeability).

From an experimental point of view, the CFP method variation used for characterization of MF-particle filtration membranes in GLDP, which is very similar to LLDP. The sample was first saturated in a highly wetting liquid that had low surface tension, low vapor pressure, and presents chemically low interaction with membrane material. Perfluoro halogenated compounds are the most successful liquids that meet such requirements and accordingly, most GLDP manufacturers usually supply

these liquids (e.g., Porofil®, from Quantachrome Inst., USA;   Silwick, Porewick, and Galwick, from PMI, USA, with the latest based on different configurations of Fluorinert®, 3M, France) to their customers [12]. In this study, the wetting liquid was fluorocarboned commercial liquid Fluorinert FC-43 (3M, France) with a surface tension value of 16 mN/m for NC samples and ultra-pure water ($\gamma$ = 72.8 mN/m) for FM, respectively. For GLDP, the displacing fluid consists of a gas steeply pressurized starting at a very low pressure and, then, the gas flow across the membrane is monitored. As the pressure progressively increases beyond the bubble point, successive pores of decreasing sizes gradually empty and contribute to gas flow $J_k$ through the membrane until all pores become empty from the wetting fluid and successive flows become proportional to the pressure. The differential air permeability contribution at each pressure step can be plotted in terms of the size of the pores yet opened by using the same Equation (1), then by obtaining the corresponding PSD, similarly to Equation (2).

### 3.2. Liquid Extrusion Porometry (LEP) Method

In order to obtain the capillary pressure curves, the same commercial device, an IFTS fluid porometer, used in previous CFP experiences was used, but now docked to an external module consisting of an appropriate sample holder and a precise analytical balance (Figure 5). The role of the CFP equipment was now to supply a stable pressure above the sample, while the balance was able to monitor the drained mass from the wetted membrane. This IFTS designed commercial setup allowed for the conversion of CFP equipment to accomplish such LEP experiments, and was adapted to characterize gas diffusion layers (GDL) integrated into polymer electrolyte membrane fuel cells (PEMFC) [22–27].

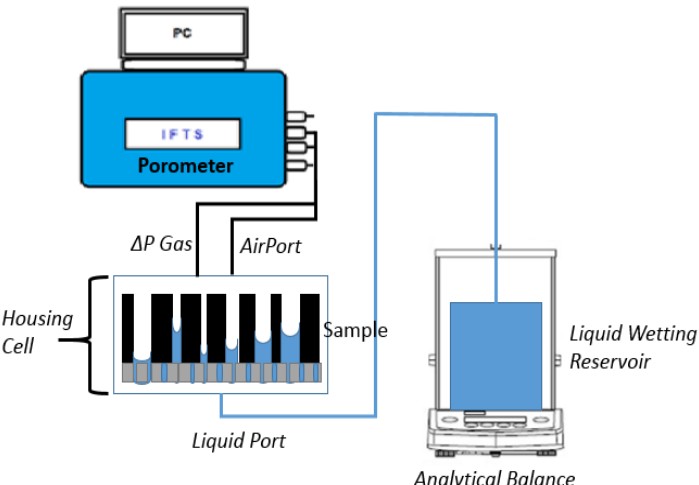

**Figure 5.** Schematic diagram of liquid extrusion porometer system setup.

The experimental device for the generation of capillary pressure curves was the drainage and further imbibition of the wetting liquid. Using the CFP porometer, the capillary pressure slowly increased and the wetting liquid flowed out of the sample onto an analytical balance (accuracy ±0.1 mg). As the system reached the equilibrium condition *(Δm/Δt~0)* at each capillary pressure, the volume (mass) of liquid drained is recorded before the pressure is again increased. Once the maximum desired pressure has been reached, the capillary pressure is slowly relieved and the wetting fluid flows back into the sample. Using this data, a capillary pressure curve (and drainage) can be generated.

The capillary pressure can be defined from the next equation [26].

$$Pc = P \text{ (wetting liquid)} - P \text{ (non-wetting fluid)} = \varrho gh + P_{Atm} - P_G \qquad (3)$$

where the gas pressure $P_G$ *(Pa)* is controlled by the porometer above the sample, and *h (cm)* is the distance between the liquid reservoir level and the sample surface, which was roughly 5 cm. Therefore, during the experiment, there was a small column of liquid open to barometric pressure $P_{Atm}$ *(Pa)* and acted on the surface of the liquid reservoir. Concerning the role of the capillary barrier needed to properly apply LEP, a hydrophilic membrane (Sartorius, 0.2 μm nominal pore size, made of CN and having Ø = 25 mm) was placed below the measured sample and performed this role as long as its pores were clearly smaller than those of the membrane to be analyzed. The main characteristics required of the wetting liquid used in the LEP experiments were very similar to those used for CFP. In addition, ideal flow properties (low viscosity) and low propensity for evaporation at room temperature are required. Evaporation is not desirable as it can lead to pores in both the sample and the hydrophilic barrier membrane becoming not wet, therefore allowing for air to breakthrough.

In the case of the MF Sartorius membrane analyzed, the mixture consisted of the aqueous phase (see Section 3.1) from the water/isobutanol mixing. This liquid was successfully used as a wetting liquid with a surface tension value of 29.6 mN/m. Ultra-pure water was used as the wetting liquid for FM to obtain a better resolution with a low pressure manometer.

The plot of contributions to total mass drained was considered only for positive capillary pressure of the drained liquid for each opened pore and converted into PSD, in terms of normalized differential mass drained divided by differential diameter *dm/dD (mg/μm) (%)*.

$$\Delta m / \Delta d)_k(\%) = \frac{(\Delta m / \Delta d)_k}{\sum (\Delta m / \Delta d)_k} 100 \tag{4}$$

The pore sizes obtained by LEP were the average $d_p$ *(m)* from the gas capillary pressure curve.

## 4. Results and Discussion

### 4.1. Flux vs. Pressure for the Flow Capillary Porometry Technique

The PSD of Anopore (0.02 μm), PET, NC Sartorius (5 μm), and Filter Mask were obtained via LLDP/GLDP, depending on the size of the pore population for each membrane. Flow capillary porometry, as previously mentioned, is based on the effluent (i.e., flux-pressure) curve obtained when consecutive pores of the membranes are successively subjected to the flow of the displacing fluid (i.e., aqueous phase for UF filters and air for MF and particle filtration samples). The resulting curve is expected to be S-shaped, with the maximum slope corresponding to the moment all the pores are opened to flow so that the permeability becomes constant (for LLDP, as can be clearly seen in Figure 6) and slightly changes with pressure (for GLDP, corresponding to a compressible displacing fluid, Figure 7).

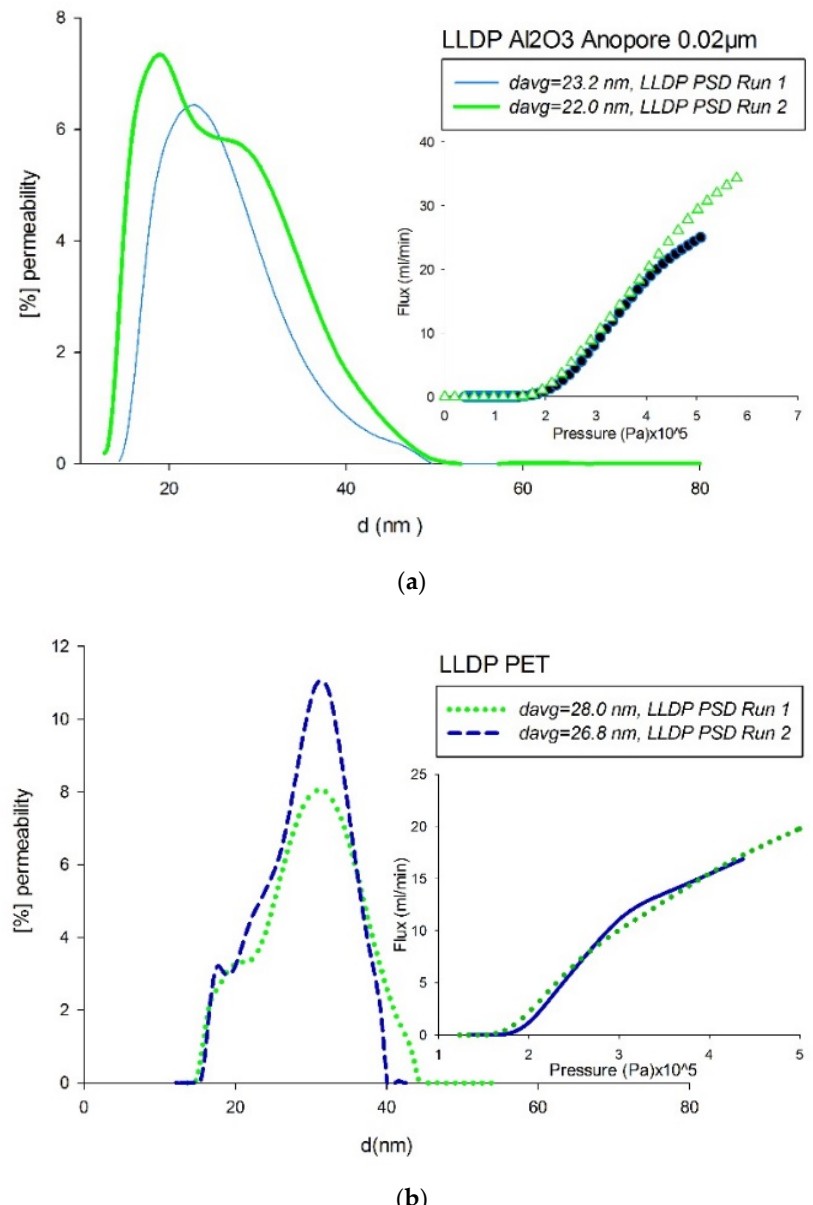

(**a**)

(**b**)

**Figure 6.** (**a**) Pore-size distributions (PSDs) of the Al$_2$O$_3$ membrane and (**b**) PET membrane as obtained using Liquid Liquid Displacement Porometry

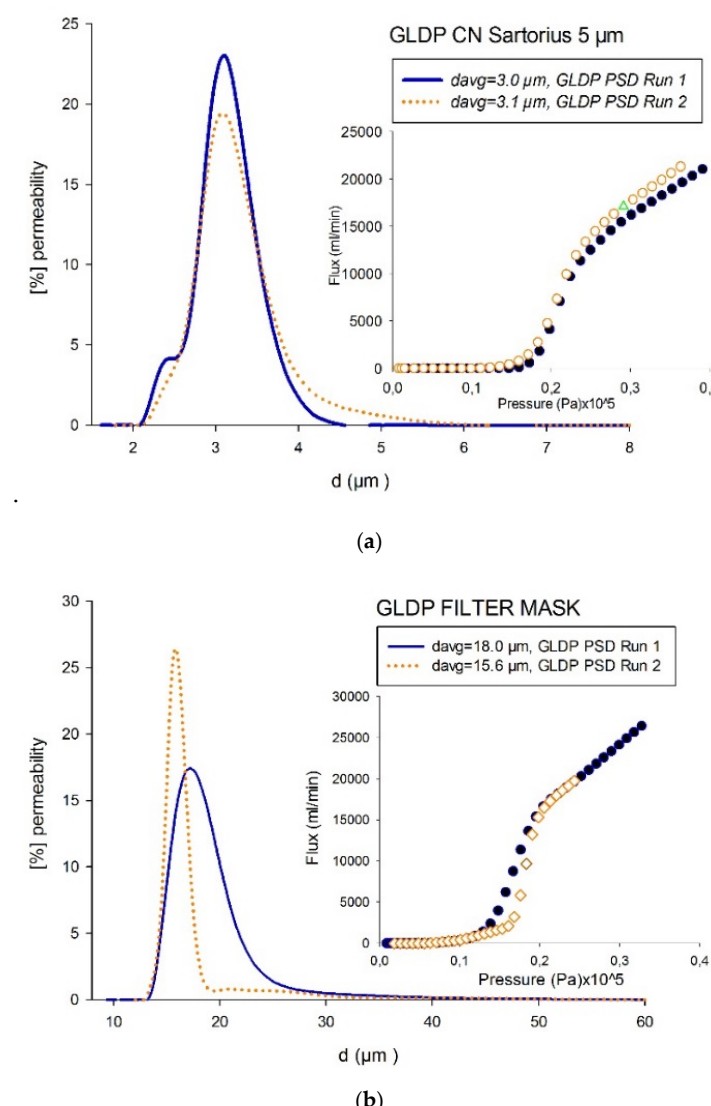

(**a**)

(**b**)

**Figure 7.** (**a**) Pore-size distributions (PSDs) of the NC membrane and (**b**) face mask as obtained using Gas Liquid Displacement Porometry (GLDP).

*4.2. Drained Mass vs. Capillary Pressure for the Liquid Extrusion Porometry Technique*

Similarly, PSD was characterized via LEP based on the extrusion mass of the wetting liquid by air (non-wetting fluid) for the case of NC Sartorius (5 μm) and FM (it must be considered that LEP only applies to MF and more opened filters, while for UF it needs too high pressures). As previously commented on in the LEP technique section, it was performed using a FFP porometer connected to the gas port of the housing cell. The porometer compresses the gas above the sample, thereby changing the capillary pressure. After each change in gas pressure and therefore capillary pressure, the mass of liquid on the balance was monitored and the corresponding changes recorded. The system was held at a constant gas pressure until the wetting fluid (water saturated in isobutanol) mass reading on the balance became stable. The resulting curve was an increasing one (similar to those presented in Figure 8), until all of the contained mass of the liquid was drained or extruded from the media and an almost constant plateau was achieved.

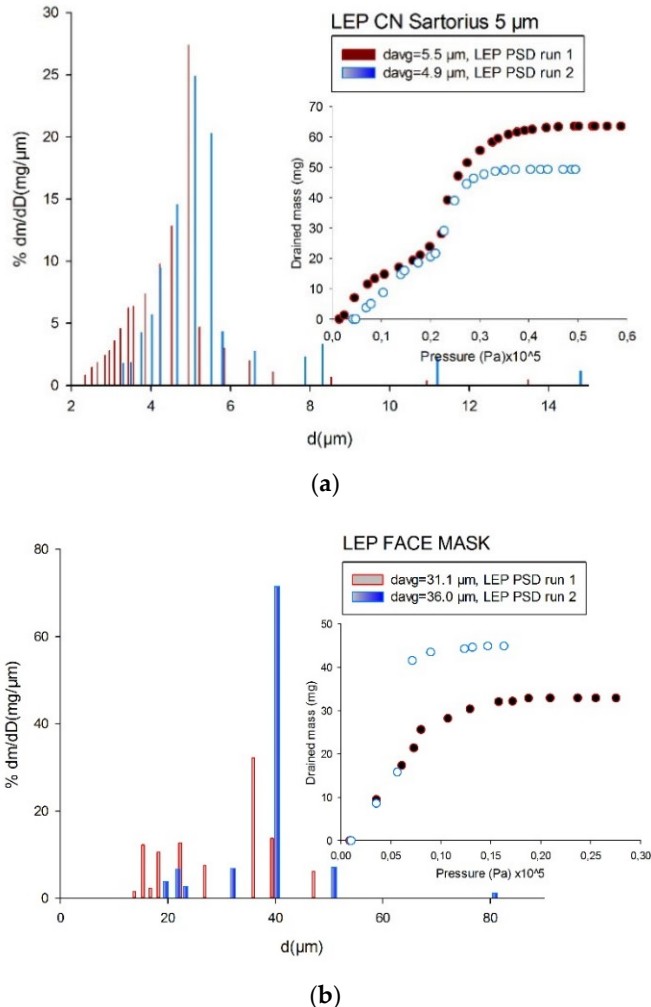

**Figure 8.** (**a**) Pore-size distributions (PSDs) of the CN membrane and (**b**) face mask as obtained using the Liquid Extrusion Porometry (LEP) technique.

### 4.3. Comparison of Pore-Size Distributions (PSDs)

The FCP and LEP techniques provide flow-based and drained mass-based diameters, respectively, and both are based on the same Young–Laplace equation to estimate the PSD. Obviously, the experimental behavior of each technique and, consequently, the resulting PSD could be slightly different, depending on the interaction of the displacement or even extrusion liquid with each sample material, but also depending on the actual microstructure or pore connectivity of the membrane, which varies from one type to another. The aim of this study was to compare such results directly in the possible range (MF to particle filtration). For each membrane and technique, two repeated runs were carried out and the results were averaged.

Figure 6a,b displays the PSDs and porometry run curves for the Anopore and PET membranes obtained via the LLDP technique, respectively. The pore diameters measured were in the range of 15–50 nm and the average pore diameter $d_{avg}$ was 22.6 ± 2.6% nm for Anopore, while for the case of the PET sample, the pore diameters measured were in a narrower range of 17–40 nm with a distinctive peak pore diameter of 31 nm and a permeability-based $d_{avg}$ of 27.4 ± 2.1%. Remember that $d_{avg}$ can be considered as a reliable estimation of mean pore size that allows for the size of particles typically retained by the filter to be estimated. Then, key observations for these two samples are that: *(i)* the repeated PSDs were similar for each sample, indicating good reproducibility and thereby reliability;

and *(ii)* the $d_{avg}$ values given by the LLDP method and nominal pore size were in good agreement (when nominal sizes were available).

Figure 7a shows the PSDs of the CN Sartorius (5 μm) membrane as obtained from GLDP. The $d_{avg}$ value was 3.11 μm with good reproducibility (error <1%). In the case of the FM filter (Figure 7b), GLDP gave an average for pore diameter of 16.8 μm; in this case, showing a sensibly higher discrepancy between both runs (±7.0%), which is also clear from both runs in Figure 7.

Finally, the results from the LEP technique for both membranes are shown in Figure 8 where the LEP technique gave a higher variability than GLDP. Therefore, for the case of the CN membrane, the $d_{avg}$ value was 5.2 ± 5.7% μm, showing good reproducibility but clearly lower than that of GLDP and a slightly higher value when compared with GLDP.

Notably, in contrast to the previous sample, the discrepancy between the GLDP and LEP results was clearly significant in the face mask. The results obtained for LEP also presented some variability with a mean value of $d_{avg}$ = 33.5 ± 7.1% μm. This sample to sample variability, which was higher for FM than for any other membrane studied (as can be seen in Table 2) can be attributed to the different fabrication method and the not-so strict control of the fabrication parameters. However, surely more important is that the different behavior of the FM filter (higher sample to sample variability) when compared with the CN membrane is clearly related with the inner microstructure of the FM fibrous medium, which can hardly be described as a simple bundle of parallel cylindrical pores [8,9]. Particularly high is the discrepancy found when comparing the mean pore sizes for this FM filter from both techniques, resulting in a difference as large as 33.5 compared to 16.8 μm. These differences can be related with a much broader PSD from 10 to 80 μm, as approximately obtained for the LEP technique, which could be explained due a small contribution of extruded mass but enough to be registered at lower pressures where GLDP did not take account of any flow at this low pressure.

Finally, Table 2 summarizes the mean pore size values found for each membrane analyzed along with the experimental errors. As previously commented, the results for the UF membranes were obtained from LLDP, while the GLDP and LEP outputs were compared in the case of MF and particle filtration. Finally, the obtained values were compared, where possible, with nominal values as supplied by manufacturers, with a reasonable agreement for the Anopore membrane and better for LEP than GLDP in the case of the CN membrane.

**Table 2.** Mean pore diameters ($d_{avg}$) obtained via FCP and LEP for the four membranes tested.

| Membrane | Nominal $d_{avg}$ (μm) | Young-Laplace Technique | $D_{avg}$ (μM) Flow-Based Or Mass Based | Wetting Liquid |
|---|---|---|---|---|
| Al$_2$O$_3$ | 0.02 | LLDP | 0.0226 ± 2.6% | Alcoholic Phase |
| PET | Not available | LLDP | 0.0274 ± 2.1% | Alcoholic Phase |
| NC | 5 | GLDP | 3.1 ± < 1% | FC-43 |
| | | LEP | 5.2 ± 5.7% | Water Rich Phase |
| FM | Not available | GLDP | 16.8 ± 7.1% | Ultra-Pure Water |
| | | LEP | 33.5 ± 7.0% | Ultra-Pure Water |

## 5. Conclusions

Four series of three polymeric and one inorganic flat membranes were analyzed using a precise, accurate, and fast automated CFP/LEP commercial device. The device designed and marketed by IFTS is an accurate CFP porometer that has been improved by connecting it to an analytical balance and proper LEP cell. The setup was able to analyze filters ranging from UF to particle filtration membranes. Both techniques (CFP and LEP) were developed to provide information about the PSD of membranes and filter media, and make use of the same Young–Laplace equation to convert

experimental data into pore size values. The techniques differ in the role of wetting/pushing fluids in the process of emptying membrane pores from a previous filling fluid.

For two UF membranes, only CFP (by using two liquid phases, then LLDP) was used to obtain PSD information, while for bigger pores, the filter (MF and particle filtration) results from CFP and LEP were obtained and compared.

The results are very interesting, and showed a nice agreement between different runs for both techniques (slightly better in the case of CFP) and reasonable agreement was also found when comparing the CFP and LEP outputs, except that the FM filter, which is hardly considered as a bundle of parallel pores as their fibrous structure is closer to a system of highly interconnected pores. This led to a broader PSD coming from LEP than that from the procedure based on the flow-rate curve (which in turn, was not able to discriminate bigger pores not contributing to flow). As a result, the information on the PSD, and mainly, the mean pore sizes obtained from it, can be questioned and must be assessed.

In conclusion, we can assert that an accurate LEP technique measuring air–liquid capillary pressure curves of porous and non-porous structure materials has been developed, where the results are coherent with porous membranes, especially when these membranes are well resembled by a capillary pore model.

**Author Contributions:** Peinador made the measurements and Ben Aim and Calvo collaborated in the interpretation of results and paper writing. All authors have read and agreed to the published version of the manuscript.

**Funding:** This research was funded by the French "Ministère de l'Enseignement supérieur, de la Recherche et de l'Innovation" through by (Crédit Impôt de la Recherche CIR-IFTS-2020).

**Conflicts of Interest:** The authors declare no conflict of interest.

## Nomenclature

| | |
|---|---|
| $Al_2O_3$ | Alumina |
| $\Delta P$ | Differential pressure (Pa, bar) |
| $\Delta m$ | Differential mass (mg) |
| $\Delta d$ | Differential pore diameter (m) |
| $d_p$ | Pore diameter (m) |
| $d_{avg}$ | Mean pore diameter (m) |
| FM | Face mask |
| GLDP | Gas liquid displacement porometry |
| HgP | Mercury porometry |
| $h$ | Distance between the liquid reservoir level and sample surface (LEP porometry) |
| IFTS | Institut de la Filtration et des Techniques Séparatives |
| $J$ | Volume flow ($m^3$/s) |
| $L$ | Permeability ($m^3$/Pa s) |
| LEIP | Liquid intrusion/extrusion porometry |
| LEP | Liquid extrusion porometry |
| LLDP | Liquid-liquid displacement porometry |
| $m$ | Drained mass (mg) |
| MF | Micro Filtration |
| MWCO | Molecular Weight Cut Off (Da) |
| PF | Particle filtration |
| PEMFC | Polymer Electrolyte Membrane Fuel Cells |
| $P_G$ | Gas pressure (Pa) |
| $P_{Atm}$ | Atmospheric Pressure (Pa) |
| $P_c$ | Capillary pressure (Pa) |
| PET | Polyethylene membrane |
| PSD | Pore size distribution |

| UF | Ultrafiltration |
| --- | --- |

**Greek Letters**

| $\gamma$ | Interfacial or surface tension of the fluid pairs and membrane surface (mN/m) |
| --- | --- |
| p | Liquid density (Kg/m3) |
| θ | Contact angle (°) |

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
