# Peer review of "Comparison of Capillary Flow Porometry (CFP) and Liquid Extrusion Porometry (LEP) Techniques for the Characterization of Porous and Face Mask Membranes"

_applsci, doi:10.3390/app10165703_

Round 1
Reviewer 1 Report
This work is relevant and interesting for readers. The authors consider important issues in membrane research. Good, full-fledged experimental work has been done. However, I have several comments:
1) Why were the experiments limited to only two repetitions? I believe that two repetitions cannot be used to judge the convergence of the results.
2) Although a lot of experimental work has been done, the general conclusion about the methods for measuring the pore size distribution could be predicted before the experiments. I would like to see not only statements of facts from the obtained experiments, but also substantiated explanations to them, as well as recommendations for applying the methods to various systems.
Author Response
Dear Reviewer, thanks so much for your revision.
I am glad to respond to your questions.
This work is relevant and interesting for readers. The authors consider important issues in membrane research. Good, full-fledged experimental work has been done. However, I have several comments:
- Why were the experiments limited to only two repetitions? I believe that two repetitions cannot be used to judge the convergence of the results.
Yes, I agree, in our case, some membranes coming for Company A and B were limited in samples to be measured, previously It was used several to correlate under other characterization techniques, and unfortunately, only two samples from the same batch were profitable to get analyzed.
Finally, in order to get more homogeneity to the work, we choose the minimum quantity of the batched tested to harmonize getting better the presentation.
- Although a lot of experimental work has been done, the general conclusion about the methods for measuring the pore size distribution could be predicted before the experiments. I would like to see not only statements of facts from the obtained experiments, but also substantiated explanations to them, as well as recommendations for applying the methods to various systems.
In our case of MF-Particle filtration membranes has been analyzed using IFTS tabletop SEM microscopy to compare direct techniques to indirect ones,
The PET membrane was published in a comparison between LLDP vs EP in (JMS) https://doi.org/10.1016/j.memsci.2019.05.077 and finally, the Anopore Track etched sample, it is well correlated as a standard pore for calibration pursuits.
Best Regards
Rn
Reviewer 2 Report
The current manuscript entitled, "Comparison of capillary flow porometry (CFP) and liquid extrusion porometry (LEP) techniques for the characterization of porous and face mask membranes" describes the preparation of PET, CN, FM, and alumina membranes to measure the pores of the membranes. The work itself is interesting.
I would therefore, recommend the manuscript to be accepted after the following issues are resolved.
- In the introduction, the need and novelty of this work should be further clarified.
- The order of the contents needs to be revised, ( 2.1 cappillary…. 2.2 Liquid Int… 3.1 cappillary..3.2 Liquid Int…)
- In the table. 2, it can seen that different wetting solutions were used for each membrane type. However, only the FM was explained in the Methods part. Others should be explained as well.
- It would be nice if the image of the membranes used in the experiment was added.
- Please measure the SEM to directly compare the measurement results.
- Add the caption for the figure.5.
- Add the quantitative range or point (on line 358)
Author Response
Dear Reviewer,
Thanks so much for your revision
I am glad to respond to your question
The current manuscript entitled, "Comparison of capillary flow porometry (CFP) and liquid extrusion porometry (LEP) techniques for the characterization of porous and face mask membranes" describes the preparation of PET, CN, FM, and alumina membranes to measure the pores of the membranes. The work itself is interesting.
I would, therefore, recommend the manuscript to be accepted after the following issues are resolved.
- In the introduction, the need and novelty of this work should be further clarified.
Done,
- The order of the contents needs to be revised, ( 2.1 capillary…. 2.2 Liquid Int… 3.1
Done
- In the table. 2, it can see that different wetting solutions were used for each membrane type. However, only the FM was explained in the Methods part. Others should be explained as well.
Done,
- It would be nice if the image of the membranes used in the experiment was added.
Done, just only MF and Particle filtrations membranes were analyzed using SEM, UF samples resolution did not give an optimal images.
- Please measure the SEM to directly compare the measurement results.
Done
- Add the caption for the figure.5.
Done
- Add the quantitative range or point (on line 358)
- Done
Best regards
Rn